# The Influence of Biomolecule Composition on Colloidal Beer Structure

**DOI:** 10.3390/biom12010024

**Published:** 2021-12-24

**Authors:** Irina N. Gribkova, Michail N. Eliseev, Yuri D. Belkin, Maxim A. Zakharov, Olga A. Kosareva

**Affiliations:** 1All-Russian Scientific Research Institute of Brewing, Beverage and Wine Industry—Branch of V.M. Gorbatov Federal Research Center for Food Systems, 119021 Moscow, Russia; mazakharoff@mail.ru; 2Academic Department of Commodity Science and Commodity Examination, Plekhanov Russian University of Economics, 119021 Moscow, Russia; michail_eliseev@mail.ru (M.N.E.); Belkin.YD@rea.ru (Y.D.B.); 3Department of Commerce and Trading, Non-State Private Educational Institution of Higher Professional Education, Moscow University for Industry and Finance “Synergy”, 125190 Moscow, Russia; oakosareva@mail.ru

**Keywords:** beer, nitrogen compounds, polyphenol compounds, β-glucan content, fractionation, patterns

## Abstract

Recent studies have revealed an interest in the composition of beer biomolecules as a colloidal system and their influence on the formation of beer taste. The purpose of this research was to establish biochemical interactions between the biomolecules of plant-based raw materials of beer in order to understand the overall structure of beer as a complex system of bound biomolecules. Generally accepted methods of analytical research in the field of brewing, biochemistry and proteomics were used to solve the research objectives. The studies allowed us to establish the relationship between the grain and plant-based raw materials used, as well as the processing technologies and biomolecular profiles of beer. The qualitative profile of the distribution of protein compounds as a framework for the formation of a colloidal system and the role of carbohydrate dextrins and phenol compounds are given. This article provides information about the presence of biogenic compounds in the structure of beer that positively affect the functioning of the body. A critical assessment of the influence of some parameters on the completeness of beer taste by biomolecules is given. Conclusion: the conducted analytical studies allowed us to confirm the hypothesis about the nitrogen structure of beer and the relationship of other biomolecules with protein substances, and to identify the main factors affecting the distribution of biomolecules by fractions.

## 1. Introduction

The alcoholic beverage, as it is known, is a colloidal structure, with a composition formed by primary (from vegetable raw materials) or secondary biomolecules, which are the product of the vital activity of microorganisms or subsequently formed in the result of biochemical, chemical or other processes of a different nature during production. The combination of primary and secondary organic compounds affects the taste perception of the consumer audience and generates demand in the goods market. Beer may also be attributed to beverages with a complex colloidal structure formed by organic biomolecules of various molecular weights interconnected by hydrogen, covalent, disulfide and other bonds [1,2,3].

It should be recalled that the term “colloid” is of Greek origin and forms the stem of the word “kola” (Greek), which means “glue” [4]. The term has been used since 1862 for the gradation of liquid objects that are colloidal dispersions. In 1961, two classes of substances belonging to colloids were distinguished: colloidal crystals and colloids, differing in the rate of diffusion through biological membranes of animal and plant origin [5]. Further studies of the colloidal liquid properties allowed us to conclude that a suspension of the smallest particles in a continuous phase or a dispersion medium can be defined as a colloidal dispersion; in it, suspended particles of the biological matrix may represent separate large molecules interacting with each other, or aggregates of molecules or ions with a size range of 1–1000 nm [6].

Beer meets the requirements that are applied to colloidal systems [7]: multicomponent and multiphase (gas/liquid) composition, insolubility of substances, as well as multiple intermolecular interactions [7,8].

It is interesting to note that the classes of organic compounds involved in the formation of the sensory profile of beer and responsible for the stability of the fermented beverage are basically similar—these are nitrogen compounds, phenol compounds, carbohydrate biomolecules and some other compounds of plant-based raw materials [3]. Of course, the formation of the beer’s taste profile is versatile and depends on both primary biomolecules hydrolyzed during the process stages of wort production, and secondary biomolecules appearing at the fermentation stage as a result of biomodification in the Krebs cycle [9].

Some biomolecules of certain molecular weight, or of so-called fraction, are involved in the formation of both the taste profile and consumer characteristics of beer, while others contribute to the formation of haze, and it is important to distinguish them.

## 2. The Beer’s “Head” Biomolecules

Important consumer characteristics that are associated with the quality of beer are the quality of foam—its durability and deposition time [10]. It is noted that the colloidal size of the foam is determined by the thickness of the compound film, and not by the size of the gaseous substance during the formation of a thick foam [4]. The stability of the foam depends on the properties of the compounds forming it—liquid films of biomolecules tend to reduce their thickness and break the integrity of the surface-active film and molecules of gaseous compounds tend to diffuse and evaporate through liquid films [4].

### 2.1. The Effect of Proteins on Foaming

The foam formation and quality are closely related to protein fractions, bitter hop resins, pentosans, gum substances and other fractions of plant raw materials that form foam on the surface of carbon dioxide bubbles [11]. It is noted that it is protein biomolecules that play the main role in the formation of the foam of brewing products, and there is a differentiation: some proteins exhibit foaming properties, while others show stabilizing ones [12]. It has been shown that the foam composition is formed by lipid transfer proteins (LTP1) with a molecular weight of 9.7 kDa and 91 amino acids in the composition; protein Z has a molecular weight of 40 kDa and various hordein derivatives ranging in size from 10 to 30 kDa [13,14].

Table 1 shows the composition of the beer protein compounds.

Non-specific lipid transfer proteins (LTP1), which have been actively studied recently [17,18,19,20,21,22,23,24,25,26,27,28,29,30,31,32,33,34,35,36,37,38,39,40,41,42,43], play the primary role in foaming. In their structure, LTPs contain cysteine residues with bound disulfide residues, which stabilize the three-dimensional structure of nsLTP, consisting of four or five alpha-helices, connecting loops and an unstructured C-terminal end, and provide thermal and chemical stability of the molecule [44]. The proteins are of vegetable origin and regulate the growth functions of plants [35].

Studies have shown the uniqueness of the set of genes responsible for the reproduction of LTP protein in cereals [45,46].

Table 2 shows the total number of nsLTP gene families in the main cereal crops used in the brewing process [45]. The researchers noted differences between grain crops in the types of genes and the uniqueness of type E for dicotyledons, while barley LTPs are also deficient for types C and X. Extended genomic analysis has shown that due to the reduction in the barley population and its types, there are changes in the structural pattern of genes, as well as physiological processes associated with their expansion. The theoretical isoelectric point (PI) and molecular weight (MW) were analyzed for all the putative LTPs of barley: the average length is 132 aa (91–200 aa), with a molecular weight ranging from 9206.81 to 19,981.15 Da. The average MW LTPs of barley is 13,344 Da with an average theoretical indicator of the isoelectric point pI LTPs of 8.07, which shows the main properties of this protein sequence [45].

The structure of lipid transfer proteins varies depending on the class. It is characterized by a different cavity structure, which is able to bind to various ligand compounds and provide either stability or instability of the colloidal foam structure. It was found that the addition of a palmitic acid residue as a ligand through the terminal thiol-containing acyl-CoA during the metabolism of fatty acids and other lipids, the barley protein LTP1.1 crystallized, since the volume of the cavity changed from 39 A3 in the case of unbound LTP1.1 barley to 620 A3 for the ligand-containing one [47]. The structure of the internal cavity of LTP is determined by the type of grain crop. Experimental studies of barley and corn lipid transfer proteins (LTPS) have shown that the binding of palmitate ligands occurs in opposite orientations in the internal cavities of barley and wheat protein. It has also been shown that there are differences in the binding behavior of fatty acid ligands in corn and barley LTP in the hydrophobic cavity of LTP, which is associated with the presence of Arg46 residue that prevents lipid binding [48,49]. The mechanism of binding of the ligand caprate in the internal cavity with respect to two orientations occurs by attaching the carboxylate ligand and the arginine side chain through the salt residue of the head group.

Hop proteins, which contribute to the biochemical composition of beer at the boil stage, are characterized by sizes of 5 kDa, 15 kDa and 25 kDa for both bitter and aromatic hop varieties [50]. It has been shown that these nitrogenous compounds, along with grain proteins, are involved in the formation of foam and the body of beer [51]. However, the great value of nitrogenous hop compounds for their own growth and development, as well as the induction of individual compounds, should be noted [52,53,54]. The extraction of protein from the hop side is limited by the hop product types used in the technology and the extraction conditions, since the extracts and other processed products do not contain a large amount of organic hop compounds and are narrowly targeted. 

### 2.2. The Ligand Compounds of Proteins Involved in Foaming

The ligand compounds may be represented by bitter isoforms of α-bitter hop acids bound by covalent bonds between the carboxyl group of the asparagine residue in LTP1 molecules with the resin hydroxy group, flavonoids, phytosterols, etc. [55,56]. Due to the open ability of native LTP to bind the fluorescent probe of 8-anilino-1-naphthalenesulfonic acid (ANS) using NMR, it was possible to identify the protein cavity as a place of binding and identify the presence of a number of physiologically relevant ligands (flavonoids, cytokinins and phytosterols) by displacing ANS [55,56]. The interaction of isohumulone and protein Z with the formation of a coherent and elastic adsorbed layer inside the TLP cavity has been established, which has a positive effect on the stability of beer foam [51].

Much attention has been paid to the effect of protein Z on foam stability [57,58]. It was found that the level of the foam structure stability was affected by the hydrophobicity of the structure of protein compounds: lower surface hydrophobicity contributed to the most complete interaction with the active sites of the TLP cavity. It was found that the amount of protein Z positively correlates with the stability of beer foam, but this relationship is non-permanent [57,59].

It was found that protein Z is part of the hordein protein fraction and that good malt solubility leads to a greater release of this protein into the liquid fraction [60], which, on the other hand, causes protein Z to have an effect on the haze formation [61]. This fact is confirmed by other studies that have established the fact that the intensity of haze is associated with the content of barley malt fractions with a molecular weight of 8–14 kDa and wheat malt fractions of <7 kDa [62]. There is a fraction of protein Z with a larger molecular weight of 40 kDa and a range of pI from 5.5 to 5.8 [63]. It should be noted that protein Z is thermally stable and resistant to the enzymatic effects of degradation, and this causes its presence in finished beer [63,64]. The modification of lysine residues in the serpine molecule (protein Z4), or so-called glycation, during malt drying as a result of Maillard reaction with sugars contributes to the thermal stability of serpine, which is involved in the stabilization of foam [65]. Glycation of these two very important foam-positive proteins suggests that the crosslinking of sugars and proteins is necessary to obtain foaming proteins [66].

Note that, on the other hand, studies of the last 40 years have allowed us to assert that there is a partial similarity in the composition of protein fractions of foam and the “body” of beer, consisting of three groups of protein molecules of 40, 10 and 8 kDa (proteins and peptides) with affinity for nitrogen compounds of barley (data from Table 1) [67,68].

## 3. The Beer’s “Body” Biomolecules

The completeness of beer perception is formed by such biomolecules as non-starch polysaccharides, phenol compounds, bitter resins of hop products, melanoidins, reductones, etc.

### 3.1. The Beer’s “Body” Proteins

We mentioned earlier the interaction between proteins and monosaccharides as a result of the Maillard glycation reaction [66]. The conjugation scheme is shown in Figure 1.

Studies have shown that metal-free phosphate ions catalyze the formation of sugar-protein molecules, and the number of carbon atoms also matters for the reaction rate, since the crosslinking of fructose and lysine occurs faster compared to glucose [69]. This fact is of great importance, since it is the phosphate ions that maintain buffering capacity in the colloidal system of beer.

A critical assessment of the qualitative composition of protein molecules forming the “body” of beer is given in Table 1 and indicates some similarity in the composition of biomolecules of foam and the “body” of beer.

### 3.2. The Carbohydrates “Body” Profile

In addition to the monosaccharides present in the colloidal system of beer, other non-starch polysaccharides are important. Much attention is paid to the influence of arabinoxylans and β-glucans on the viscosity and stability of beer [70]. The viscosity was caused by the presence of dextrins of non-starch polysaccharides with a high molecular weight capable of forming viscous gels, which is associated with insufficiently complete dissolution of the malt endosperm during malting [71,72,73]. The source of concern was that during the extraction of malt arabinoxylans, ferulic acid dimers were found in their composition, which indicates a crosslinking reaction of phenol compounds and polysaccharides, causing either filtration problems or taste harmony depending on the type of molecules being crosslinked [74,75]. Crosslinking occurs through the esterification of arabinoxylans through covalent, hydrogen and Van der Waals interactions, and the resulting ferulates are involved in the physiological processes of crops [76,77,78].

Regarding non-starch polysaccharides, the effect of mannans with a molecular weight of 3744–7632 Da and 7470–26,262 Da on the viscosity and taste of beer was noted [79]. The content of mannose polymers, expressed as mannan, for beer was at the level of 120–180 mg/L and is a consequence of the use of grain raw materials and microorganisms in the process [80]. Mannans in the structure of beer can be linked with proteins and represent compounds of yeast cells—mannoproteins involved in the haze formation [81].

It has been shown that β-glucan, along with maltodextrins, enhances the saturation of taste in the mouth. The molecular weight of β-glucan in barley, malt and beer ranges from 10 to 10,000 kDa, and the molecules are conjugates with protein and phenol molecules [82,83,84]. It was noted that β-glucan dextrins with a molecular weight of up to 200 kDa positively affect the completeness of the beer taste, and molecules with a weight above 200 kDa interact with protein–polyphenol associates, increasing the beer haze [85,86,87,88]. Researchers note the primary role of the concentration of arabinoxylan and β-glucan molecules, rather than their molecular weights, for the formation of viscous solutions in beer [89]. In our opinion, this fact suggests that an increase in the concentration of non-starch polysaccharides entails an increase in the level of proteins, and carbohydrate molecules, previously inhibiting aggregation, are not able to even the equilibrium state of the nitrogen molecules associated with them.

In general, up to 23% of starch dextrin remains from carbohydrates in beer, which is not stained with iodine and is not fermented by yeast. The content of arabinoxylan and β-glucan is 2.5 and 0–5 g/L, respectively [89]. Studies performed based on determination of various fractions of polysaccharides revealed a coefficient of correlation between the content of nitrogen substances, β-glucan, viscosity and high-polymeric fraction of biomolecules of 0.27, 0.37, 0.54 and 0.15, respectively [82]. The determination coefficient calculated on the basis of the correlation coefficient as the square root of the correlation coefficient value shows the determination coefficient, taking into account the percentage of the analyzed parameters involved in the process. It turns out that the viscosity (0.73) has the greatest effect on the completeness of taste of these parameters, followed by the content of β-glucan (0.61) and the content of nitrogen substances (0.52). The model studied by the authors [82] did not take into account polyphenols, fermentable sugars, chloride ions and glycerin [90].

### 3.3. The Polyphenol Beer’s “Body” Profiles 

Plant polysaccharides containing OH groups can be conjugated or covalently bound to polyphenols [91]. The reduction–oxidation system of malt forms free hydroxyl radicals on the surface of polysaccharides that react with polyphenols. Such molecules function as antioxidants; they regulate the activity of enzymes (α-glucosidase, α-amylase, acetyl cholinesterase, tyrosinase) and have antimicrobial potential [92,93].

The triple conjugation of protein–polysaccharide–phenol is carried out by residues of amino groups, cysteine and tyrosine from the protein molecule, and in polysaccharides by a carboxyl group followed by crosslinking during the Maillard reaction [94].

Similar behavior of conjugates in a liquid medium consisting of various biomolecules allows the colloidal system to be structured. Double protein–polysaccharide conjugates contribute to an increase in solubilization due to steric repulsion by the polysaccharide of protein molecules tending to agglomeration [95].

The double conjugation of protein–phenol molecules depends on the nature of the phenol compound affecting the electrical potential (isoelectric point) of nitrogen molecules [96]. The polarity/non-polarity of phenols is important, since non-polar polyphenol molecules increase the hydrophobic charge of the protein molecule surface, which leads to sedimentation due to increased surface activity [97]. On the other hand, when proteins and polyphenols are associated, there is a change in the structural shape of protein molecules due to crosslinking, which affects the potential of the protein molecule and changes in its charge and, consequently, solubility [98]. Such representatives of phenol compounds as (−)-epigallocatechin-3-gallate, found in beer, are able to significantly reduce the solubility of protein-polyphenol conjugates in water at the level of acidity of wort and beer of pH 4 ÷ 5 [2,75,99,100,101,102]. On the other hand, the (+)-catechin and ferulic, caffeic and synapic acids found in beer, forming conjugates with polysaccharides, had antioxidant properties [74,103,104,105,106,107].

Conjugation has a great influence on the digestibility of organic biomolecules by the body, since it is double or triple conjugation that makes it possible to increase the bioavailability of useful biomolecules for the human body [108].

Since the hop products used are of great importance for beer production, it is necessary to consider the effect of bitter hop resins on interaction with other biomolecules of beer. This refers to soft resins (or bitter acids), such as α- and β-bitter acids, called humulones and lupulones, respectively, and their isoforms.

It has been found that, according to the Van der Waals laws, the interaction of bitter resins is the least. Association with carbohydrate and protein molecules occurs through hydrogen bonds through OH- or NH_4_-primary thiol groups, as well as by ionic interaction. Interaction in terms of hydrophobicity occurs through non-polar resin sites and aromatic groups of amino acids, proteins and polyphenols [109].

The most important influence of iso-α-acids in terms of biochemistry is the interaction with protein Z (serpine) with the formation of a coherent and elastic adsorbed layer, which contributes to stabilization [110]. Most likely, the interaction affects the crosslinking of the active center of the serpine protein molecule, which makes it thermally stable, and the isoacid molecule spatially structures and stabilizes the protein.

Recently, the protective effect of isohumulones in the structure of beer was revealed, which provided 90% of the reaction with 1-hydroxyethyl radicals that occur during beer storage [111].

Along with bitter resins, polyphenol hop compounds are extracted, which form a phenol profile responsible for the stability of the colloidal system of beer [112]. Thus, it was found that amino acids of a hydrophilic nature (arginine, aspartic acid) contained in wort interact with hop catechins [113] through the decarboxylation of amino acid and detaching of catechin hydroxy groups to form a biogenic catechol amine, which leads to stabilization of hop polyphenols [114]. This fact makes the process of apoptosis induction in cancer cells a little clearer by inhibiting the activity of tumor necrosis factor α (TNF-α) with the participation of biogenic amine [115]. Apoptosis is caused by mitochondrial processes, as well as nuclear condensation, activation of caspase-3 and breakage of poly (ADP-ribose) polymerase under the influence of catechols on the human body [116,117].

It should be noted that hop contains about 2.821 mg/kg of (+)-catechins, and malt contains up to 16 mg/kg [112,118]. However, researchers claim that the distribution of existing polyphenols by raw material is nonproportional—80% of polyphenol compounds are of grain origin and 20% are of hop origin [112], and this is due to the degree of conjugation with other compounds in the plant matrix.

### 3.4. The Profiles of Other Beers’ “Body” Compounds

The chloride ions and glycerin mentioned earlier in the authors’ study [90] were attributed to substances that affect the fullness of the feeling of the beer “body”. The level of chlorine ions depends on the quality of processed water and raw materials. Thus, the chlorine level was stated in the range of 24–100 mg/L, depending on the country of origin [119]. The control and relevance of this ion is based on the ability of the Cl ion during processing with water to react with compounds and form trihalomethanes, haloacetic acids, haloacetonitriles, haloketones, chloral hydrate or chloropicrin, which are carcinogens [120,121]. Chloride ions can be introduced exogenously to regulate the pH of the mash, but at concentrations above 0.59 mg/L, they can lower the pH and suppress fermentation because they are included in the transport mechanism of intermembrane penetration of nutrients into the cell and thus can inhibit yeast growth [120], as well as cause toxic effects [120,121].

During beer consumption, alcohol affects the function of neural membrane receptors, and it interacts with various receptors of amino acid neurotransmitters (γ-aminobutyric acid and glutamate). Subunits of protein molecules (neurotransmitters) arranged in the form of a channel, when exposed to aclogol, are bound to receptors and begin to function, transporting small ions (Na^+^, Ca^2+^, Cl^−^) into the cell, and the type of ion passed is determined by the nature of the amino acid of the receptor [122,123,124]. This can explain the small concentrations of chloride ions sufficient to feel a salty taste.

At moderate concentrations, chloride ions cause sensations of a rounded malt aroma and a softer hop flavor [90]. The dissolved ions of the Cl- and OH-groups of phenol molecules, in particular catechins, which affect the rough hop taste of beer, interact with each other, and this leads to the incorporation of halogen groups into the phenol molecule; the hydrophobic character of the benzene ring of phenol increases with a simultaneous decrease in solubility, the concentration of phenols decreases, and therefore, the taste is rounded [125].

The components responsible for the malt flavor of beer are (E)-β-damascenone, 2-acetyl-1-pyrroline, methional, 2-ethyl-3,5-dimethylpyrazine and 4-hydroxy-2,5-dimethylfuran-3(2H)-on [126]. Other researchers also claim the influence of methional, which gives beer the aroma of boiled potatoes, 3-methylbutanal, which is responsible for the cocoa-like tone, (E)-β-damascenone, which gives an apple-like, jam-like flavor, 5-ethyl-3-hydroxy-4-methyl-2(5H)-furanone confers notes of curry, and phenyl acetaldehyde, which has floral honey aromas [127]. Methional is formed at the fermentation stage when methionine is reduced by yeast enzymes. 2-acetyl-1-pyrroline, 2-ethyl-3,5-dimethylpyrazine, 4-hydroxy-2,5-dimethylfuran-3(2H)-on and phenyl acetaldehyde are of malt origin and are formed at the stage of malt drying, mashing grain products and boiling wort with hops, that is, thermal reactions [128,129,130].

The compound of the essential oil (E)-β-damascenone, or norisoprenoid, has a completely unexplained origin. It is assumed that this aromatic compound may be formed as a result of the degradation of the carotenoid neoxanthin; on the other hand, it has been suggested that it may be a product of the conversion of β-D-glucoside 3-hydroxy-β-damascone; that is, it may be of hop origin [131,132].

Thus, it is necessary to consider the influence of Cl ions already at the stage of mashing, hopping or fermentation when discussing the effect of this anion on the malt flavor of beer. In our opinion, the mechanism of amino acid conversion, during which the above compounds can be formed in the presence of Cl ions, depends on the effect of this anion on the active transport of nutrients into and out of the cell of the yeast organism [122].

Researchers note a synergistic effect with an increase in the concentration of (E)-β-damascenone on the aroma enhancement by norisoprenoids [133]. The influence of the colloidal matrix of beverages is assumed. Therefore, it is difficult to establish the exact mechanism of the chloride ions’ influence on the change in the concentration of damascenone in beer.

A lot of work has been conducted on the contribution of glycerin to the fullness of the beer taste. The presence of glycerin causes additional viscosity and density of beer, which balances the acute perception of the beer taste due to the contribution of alcohol to the sensory profile [134].

The effect of fat-like substances on the formation of the colloidal system of beer is ambiguous and is caused by the nature of lipids. Complex lipid-like compounds’ phytosterols, which include β-sitosterol and homologues (stigmasterol, campesterol and brassicasterol) of barley and hops, as well as yeast ergosterol, are formed from lanosterol [135,136]. The biosynthesis of phytosterols has precursors in the form of acetic and mevalonic acids, which are converted in the process of fermentation to squalene with varying degrees of cyclization and conjugation [137].

Phytosterols of cereals are localized throughout the grain and can be conjugated with starch dextrins or other non-starch polysaccharides [138]. The level of phytosterols in beer depends on the raw materials used in the process. Thus, for light beer prepared on the basis of pale brewer’s malt, the level of campesterol was 21.0 μg/L, stigmasterol—11.7 μg/L. For light barley malt beer, the level of the same phytosterols was 10.4 and 9.7 mcg/L, respectively, for dark—29.3 and 25.2 μg/L, respectively, and for wheat—16.8 and 15.7 μg/L, respectively [139]. The level of yeast ergosterol in unfiltered beer was 0.08–061 mcg/L, and it was absent in filtered beer [140]. The level of hop phytosterols was about 140 mg/100 g [141].

The issue regarding phytosterols is considered in connection with their chemical capabilities to integrate into protein molecules and frame the structure of beer foam.

Studies on the effect of various lipid fractions on foam stability are presented. Low molecular fractions of lipids (triglycerides, phospholipids) reduce the surface tension of the liquid film around the gas bubbles, which leads to the collapse of foam bubbles [142]. Conversely, an increase in the hydrophobicity of the surface of protein foam molecules, namely amino acid residues, thiol groups, cations and anions, carbohydrates and lipids, contributes to the stabilization of the foam structure [143,144]. During the interaction of LTPs and lipid molecules of complex structure, such as phytosterols, Van der Waals interactions occur between the lipid and protein molecules [145]. On the other hand, opinions have been expressed that carboxylate groups in phytosterols can react with cysteine molecules in the LTP1 molecule [146].

## 4. Conclusions

The high molecular profile of cereal-based beverages is usually determined by polymer compounds (proteins, polysaccharides and polyphenols) and their progress in depolymerization during processing [82,147,148].

Critical studies of the composition of beer biomolecules in relation to foam and “body” formation allowed us to draw a conclusion about the protein-frame structure of beer [15,16,17,18,19,20,21,22,23,24,25,26,27,28,29,30,31,32,33,34,35,36,37,38,39,40,41,42,43]. Due to the complex structure of the protein molecule and the presence of reactant groups in the amino acids contained in such a molecule, it becomes possible to establish a colloidal equilibrium in the liquid/gas system. Analytical analysis made it possible to establish the dependence on the raw material regarding the protein profile of compounds that form the consumer properties of beer as a beverage [60,61,62,63,64,65,66] and only partially on the process (at the stage of boiling wort with hops) [109,110]. The achievements of researchers in the field of deciphering the structural conformation of lipid transfer proteins allows us to take the following steps towards identifying the composition of beer, since the structure of protein biomolecules varies depending on the type of grain raw materials used [47,48,49].

Polyphenols, sugars and reducing substances, bitter resins, etc., accompanying protein substances play the role of accompanying compounds in the formation of a colloidal structure, supporting the spatial structure of protein molecules, the so-called “buffering capacity” both with respect to foam proteins and other protein structures [74,75,76,77,78,79,80,81,82,83,84,85,86,87,88,89,99,100,101,102,103,104,105,106,107].

However, the phenol profile has a much more significant effect on the structure and properties of the colloidal system of beer than other participants of the “system”. The degree of stability of a protein molecule in a colloidal beer solution depends on the physical, specific and quantitative characteristics of phenols. In other words, it can be said that the type and concentration of the phenol compound has a positive or negative effect on the composition of the fermented beverage [1,2,3,103,104,105,106,107].

The interaction of phytosterols with respect to the structuring of LTPs1 remains an open question; that is, the functions of phytosterols have not been fully studied.

Thus, the equilibrium colloidal system is supported by various substances and its state depends on various raw materials and process parameters.

## Figures and Tables

**Figure 1 biomolecules-12-00024-f001:**
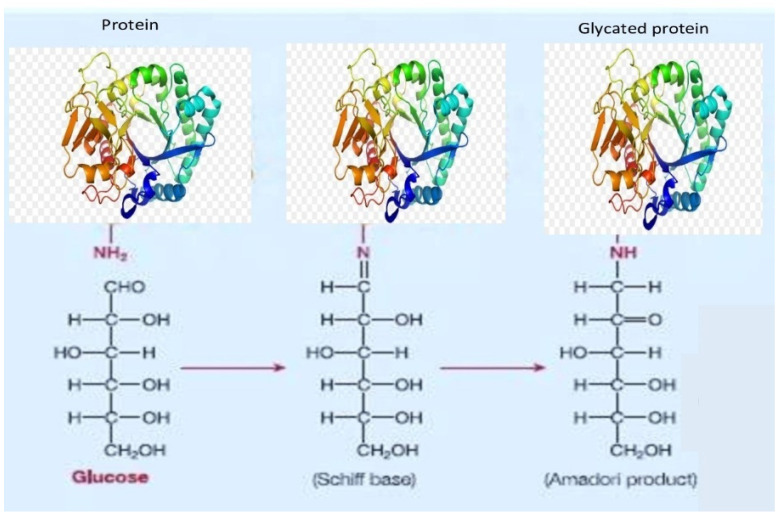
The conjugation process of proteins and sugars.

**Table 1 biomolecules-12-00024-t001:** The barley-malt beer characteristics of protein biomolecules.

Protein Fractions	Weight, Da		References
	Beer	Beer Foam	
- α/β-gliadin	-	-	[15,16]
- serpin-Z4	43,220.6 43,277.1 *	- 43,277.1 *	
- T06183 serpin	-	42,821.1 *	
- barley protein Z homolog	-	31,114.3 *	
- LTPs	12,301.4	-	
	9695.9 *	9695.9 *	
- LTP precursor 1 - LTP 7a2b	- -	14,208.4 * 12,330.5 *	
- prolamins	40,549.4	-	
- C-hordtin	50,786.0	-	
- D–hordein	75,043.0 *	-	
- D-hordein precursor			
-γ-hordein-3	33,189.1	75,043.0 *	
- γ-hordein-1	34,737.3	-	
- horgein B3	6196.4	3018.2 *	
- hotdein gomolog	-	30,182.2 *	
- hordein B	34,501.9	-	
- hordoindoline-a	16,544.3	-	
- hordoindoline b	16,126.9	-	
- β-hordothionin	14,603.2	-	
- calmodulin	16,831.8	-	
- calreticulin	-	47,038.3 *	
- dehydrin DHN3	16,162.6	-	
- γ-thionin	8931.3	-	
- 1-Cys peroxiredoxin PER1	23,963.7	-	
- globulin Beg-1 precursor	72,253.2 *	-	
-ubiquitin/ribosomal protein S27a.2 *	17,671.4 *	-	
- enzymes			
- α-amylase	15,499.9	-	
- β-amylase	59,647.9	-	
- endochitinase	33,402.8	-	
lipoxygenase	96,749.3	-	
-glyceraldehyde-3-phosphate dehydrogenase 2, cytosolic (Fragment)	33,236.1	-	
- glyceraldehyde-3-phosphate dehydrogenase 1, cytosolic	36,514.0	-	
trypsin inhibitor CME precursor	16,136.6 *	16,136.6 *	
betaine-aldehyde dehydrogenase (BADH)	54,290.2 *	-	
- metallothionein	7530.5	-	
- trypsin inhibitor CME	16,135.8	16,171.8 *	
- α-amylase inhibitor BDAI-1	16,429.5	-	
- α-amylase/trypsin inhibitor CMA	15,429.9	15,463.5*	
- α-amylaseInhibitor BDAI-I precursor *	16,430.0 *	15,817.0 *	
- α-amylase/subtilisin inhibitor	22,164.1	-	
- α-amylase/trypsin inhibitor CMd	18,525.8	-	
- α-amylase/trypsin inhibitor CMb precursor *	16,527.1 *	16,527.1 *	
- α-amylase inhibitor BDAI-1	16,429.5	16,430.1 *	
- subtilisin-chymotrypsin inhibitor C1-1A	-	8883.2 *	
starch synthase, chloroplastic/amyloplastic	87,474.8	-	
serine/threonine-protein kinase	92,872.2	-	
- ascorbate peroxidase	27,639.6	-	
Ribosonal: glycine rich protein, RNA binding protein	16,798.8	-	
embryo globulin	72,253.2	-	
barley mRNA for B1-hordein (fragment)	27,682.1	-	
ABA-inducible protein PHV A1	21,820.0	-	
putative late embryogenesis abundant protein	52,220.9	-	
glycine rich protein, RNA binding protein	16,798.8	-	
pathogenesis-related protein PRB1-3	17,697.0	-	
glutamyl-tRNA synthetase	61,863.0 *	61,863.7 *	
DNA K-type molecular chaperone HSP70	67,016.8 *	-	
RNA-dependent RNA polymerase P1-P2 fusion protein	98,746.7 *	-	

*- data from 16 issue.

**Table 2 biomolecules-12-00024-t002:** Numbers of nsLTP genes in different species.

Species	Total Samples Number	Type 1	Type 2	Type C	Type D	Type E	Type G	Type x
Hordeum vulgare	40	16	5	0	11	0	8	0
Oryza sativa	77	18	13	2	14	0	27	3
Zea mays	63	8	9	2	15	0	26	3

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
