# Peer review of "The Influence of Biomolecule Composition on Colloidal Beer Structure"

_biomolecules, 2021, doi:10.3390/biom12010024_

Round 1

Reviewer 1 Report

The authors have attempted to review compounds in beer that contribute to all the parameters of beer. There are 1000s of compounds in beer and these change according to the many beer styles and many processing techniques, so this review is very superficial. I think the authors should decide on a specific and more narrow topic from the section they have written and expand more. This review could be turned in several reviews with more detail in each. 

With the new trends in haze IPA, hoppy lagers, sour beers and low/no alcohol beers, the review really only considers the old style clear ales and lagers. One of the big trends is dry hopping but that can result in a problem called hop creep. There is no mention of this process or the molecules involved. 

What do the authors mean by 'vegetable'? Do they mean the plant-based raw materials? 

L22 The authors mention the 'hypothesis" What was the hypothesis? What was the hypothesis? Is this hypotheses by other people? And why only consider nitrogen-based molecules? If the review was just an impact on nitrogen-based molecules then the authors could make an important contribution.  

In some sections the authors discuss the molecular size or content of polymers and or molecules but structural confirmation is probably more important than the amount of a molecule. 

Author Response

The authors express their deep gratitude for reviewing the article materials

Reviewer 2 Report

The authors wrote an interesting review about biomolecules that affect beer colloidal stability.

My remarks:

The title is not clear, please simplify... perhaps to say: The influence of biomolecules composition on colloidal beer structure

Table 1: please uniform the decimal space, either use commas or dots. Also, please explain the star designated to some of the numbers. 

Line 186: the unit dm3 should be corrected to /dm3

Line 270: please uniform the values and use decimal dot to provide valid data (not sure if you meant 2.821 mg/kg or 2,821 mg/kg) 

Author Response

The authors are grateful for the review of the article materials

Round 2

Reviewer 1 Report

I still feel the content is too light weight and too broad to be such a short review. If the review is looking at nitrogenous based relationships to beer quality, the authors have missed a significant discussion on proline. It abundance in hordein and risks of hordein/polyphenols hazes, the assimilation of proline during fermentation (and yeast does assimilate proline under the right conditions) and contribution of hop proteins and amino acids during boiling and fermentation. Total FAN going into the fermentation is more than the FAN coming from malt. And FAN is not just a measure of free amino acids. These areas need some comments if the authors are covering the nitrogenous based affects on beer..

Author Response

We express our gratitude to to the reviewer for to the attential to the article. Most of the proposed corrections refer to private remarks, in the opinion of the authors, distracts from the mining of the article. The main term of the article - relationships between main biomolecules (nitrogen not amino acides, carbohydrates, phenols and others), which form the colloidal structure of beer, without touching on the turbidity topic in ditail   
